# The Effects of a Beauty Program on Self-Perception of Aging and Depression among Community-Dwelling Older Adults in an Agricultural Area in Taiwan

**DOI:** 10.3390/healthcare11101377

**Published:** 2023-05-10

**Authors:** Ya-Ling Wu, Shan-Ru Chao

**Affiliations:** 1Graduate Institute of Technological and Vocational Education, National Pingtung University of Science & Technology, Pingtung 912, Taiwan; 2Department of Social Work, National Pingtung University of Science & Technology, Pingtung 912, Taiwan; sw7735@yahoo.com.tw

**Keywords:** older adults, rural area, beauty program, beauty therapy, self-perception of aging, depression, health promotion

## Abstract

This study aimed to investigate the effects of a beauty program on the self-perception of aging and depression among the community-dwelling older adults in an agricultural area in Taiwan. Twenty-nine older adults aged 65 and above in one agricultural community care center completed the program. Based on cosmetic therapy, the beauty program consisted of 13 sessions focused on facial skin care, make-up application, and massage with essential oils. Each 90 min session of the program was conducted in groups once a week for 13 weeks. This study applied the mixed methods approach, and data were gathered through questionnaire surveys, interviews, and observation. Before and after the beauty program, the elderly individuals’ self-perceptions of aging and depression were assessed using the Attitudes towards Old People Scale (ATOPS) and Taiwanese Depression Questionnaire (TDQ), respectively. The participants’ ATOPS scores after the program were significantly higher than those examined before the program (*p* < 0.001), and their TDQ scores were significantly lower than those before the program (*p* < 0.001). Additionally, the participants’ body images were improved, the participants disrupted their stereotypes about makeup, and they were willing to gradually maintain their appearance. Overall, the beauty program was effective for enhancing the self-perceptions of aging and reducing depression in older adults in rural Taiwan. Further research with a larger population of older individuals, male older adults, or frail older adults is needed to examine the specific effects of the beauty program.

## 1. Introduction

As the population of older adults worldwide continues to grow, international organizations, governments, scholars, and practitioners across the world are promoting the health of older adults to help them live better lives while reducing the burden on care services for older people [1].

Due to the deterioration of their physical health, discontinuity of their social lives, changes in appearance, and degradation of cognitive function, older adults are usually unable to live normal or independent lives, which significantly affects their psychological health and causes low self-esteem or even depression [2,3]. Many scholars emphasize understanding older adults’ aging experiences according to the evaluation of objective indicators, such as physical health, social engagement, and safety, instead of examining their personal perceptions based on their subjective feelings [4]. In fact, even if older individuals are in good health and remain actively socially engaged, they might still feel that they have a poor quality of life [5]. Agism, which refers to discrimination, prejudice, or stereotyping towards individuals or groups due to their age, particularly towards older people, affects how older people are viewed and how they see themselves; it can also result in older people being discriminated against and in their wellbeing being damaged [6]. Therefore, to comprehensively understand the psychological processes of aging among older adults, it is important to explore their diverse life experiences from a subjective perspective [1,7,8]. How older adults perceive their own aging process is their subjective aging experience [9]. The self-perception of aging, which includes the perceptions, attitudes, and beliefs held by older adults about their aging process, is based on the assessment of their late life experiences [10,11]. Generally, older adults’ self-perceptions of aging affect their physical functioning and self-efficacy, and further influences quality of life and life satisfaction [8,12]. Hence, an older adult’s self-perception of aging, which is part of their self-concept, self-image, and empowerment, can reveal the quality of their late life experiences [9,13].

Additionally, many elderly people suffer from depression due to the deterioration of their physical and psychological functions. For the aging population, depression may cause disabilities, which can interfere with the treatment of chronic diseases and increase medical costs [14]. Elderly people who suffer from depression may experience physical symptoms due to psychological discomfort, making them seek repeated medical treatment or, in many cases, even fail to seek clinical support [15]. Depression is an important risk factor for suicide among older adults. The suicide rate of older people is twice that of younger generations, and 80% of older adults who commit suicide suffer from depression [16]. Therefore, improving quality of life by reducing depression is an important issue regarding health promotion for older adults.

In recent years, health promotion activities for older individuals launched in many countries have focused on physical exercise, music therapy, board games, gardening, animal therapy, and so on [17,18,19,20]. However, activities for delaying and preventing the functional degradation of the older population should be varied and offer many opportunities for empowerment [13]. In this way, older adults’ motivation to participate in health promotion activities can be enhanced and continued. With the increase in average life expectancy, people have begun to desire to maintain a youthful appearance rather than remaining physically healthy [21]. Moreover, some studies have suggested that older adults’ body image could be essentially detrimental to their wellbeing [22]. Studies have also confirmed that proper cosmetic intervention in clinical treatment can improve both the quality of life and psychological health of cancer patients [23]. Therefore, beauty therapy, or cosmetic therapy, has garnered academic attention in the medical and long-term care fields [24,25]. Cosmetic therapy, a type of nondrug therapy, is an intervention that involves providing or performing cosmetic practices with individuals, such as facial and body skin care, makeup application, massage, and aromatherapy, to promote their physical and psychological well-being. Based on cosmetic therapy, beauty programs designed by professionals aim to promote individuals’ physical and psychological health through a series of cosmetic interventions [3,24].

Recently, Korean and Japanese researchers have explored improving the cognitive functions, interpersonal relationships, life satisfaction, and emotions of older adults through beauty programs. Studies have shown that beauty programs can significantly facilitate cognitive functions in older adults, including orientation, attention, and arithmetic abilities, immediate and short-term memory, language ability, and visual spatial abilities [3,26,27]. Additionally, some studies have suggested that older women who participate in beauty programs tend to be satisfied with their social networks and have high levels of psychological wellbeing and low levels of depression [3,24]. Although there have been a small number of studies on beauty programs for the elderly population, the protocols of these programs have not been clearly presented, and their durations were too short (ranging from four to eight weeks) to determine specific effects. Furthermore, existing studies have seldom explored the effects of beauty programs on the self-perception of aging, and they have focused on institutionalized older adults in urban areas. Compared with elderly people in urban areas, those in rural areas, who may have less contact with beauty treatments, have different reactions to beauty programs because beauty activities are embedded in unique sociocultural contexts [28]. Most studies on health promotion programs for older adults have been carried out using quasi-experimental methods without considering the thoughts and feelings of the participants in the form of questionnaires [3,26]. In contrast, critical health psychology emphasizes developing and understanding the daily lived experiences of health and illness in real sociocultural contexts rather than reducing them to scores on a questionnaire. Furthermore, critical health psychology suggests applying multiple and qualitative research methods to comprehensively and reflectively explore the wellbeing of older people. Moreover, it is imperative for health professionals to initiate effective change aimed at improving the health and resilience of a community [6,29]. Therefore, this study aimed to investigate the effects of a beauty program on the self-perception of aging and depression of older adults in an agricultural area in Taiwan by applying the mixed methods approach.

## 2. Methodology

### 2.1. Research Site and Participants

This study, with a convenience sample of older adults aged 65 and above, was conducted at a community care center in a rural town in Pingtung, Taiwan. In recent years, the Taiwanese government has built many local nonprofit care centers for elderly people to attend all kinds of activities in their communities to facilitate their physical and mental health. Over 70% of seniors aged 65 or above in Pingtung made a living by farming when they were young. Thus, agricultural work is a shared life experience of these participants.

The research was approved by the review committee of the Pingtung County Government. Following approval by the administrators of a community care center in an agricultural town in Pingtung, the researchers individually contacted the potential participants in the community center who were aged 65 and above. The recruitment criteria for the participants included being aged 65 and over, never having participated in similar beauty programs, not being diagnosed with or suspected of having dementia, and being independent in self-care.

A total of 39 participants were recruited at the community care center and written informed consent was obtained from all of the participants. Finally, of the participants, 29 completed the beauty program, and 10 withdrew due to death (*n* = 1), fear of COVID-19 infection (*n* = 5), lack of interest (*n* = 1), and injury or sickness (*n* = 3). The average age of the 29 participants who completed the intervention was 78.6 years (SD = 5.5); most of them were aged from 76 to 80 years (34.5%). Most of the participants were females (89.7%), lived with their family (84.6%), and had received primary education or less (72.4%). All of the participants were married, and 55.2% of them had a spouse who was deceased. The average attendance rate of the program was 74%.

### 2.2. Methods

The research methodology employed in this study was a mixed methods approach, to obtain a comprehensive understanding of the phenomena being studied [30]. This was an intervention study, and all of the participants participated in the intervention. The data were collected through questionnaire surveys, semi-structured interviews, and observation. Before and after the intervention, all of the participants’ self-perceptions of aging and depression were assessed using questionnaires. In the intervention, the participants’ participation in the program was observed, and after the intervention, some of the participants and the two volunteers were interviewed.

### 2.3. Intervention Protocol

The intervention lasted for the 15 weeks from 6 April (one day after the baseline) to 27 July 2022, except the 3rd week of April and the 2nd week of July because of COVID-19 lockdowns. The beauty program intervention was conducted once a week during the 13 study weeks, and each weekly session lasted for 90 min.

The beauty program targeted older adults aged 65 and above in rural areas; its aim was for elderly participants to be able to take care of their skin, apply simple make-up on their own, perform self-massage with essential oils, have a positive self-perception of aging, and maintain positive emotions. The beauty program consisted of 13 sessions involved self-skin care, make-up application, and body massage with essential oils. The program was developed by 3 licensed beauticians, 1 registered nurse, 1 occupational therapist, and 1 health promotion expert; it was then verified by 1 licensed beautician, 2 beauty majors with master’s degrees, and 1 specialist in health promotion with a doctoral degree. The protocol of the program is shown in Table 1.

The beauty program was conducted by an experienced beauty training instructor with the assistance of the two assistants in this study and two community care center volunteers who had completed training by the instructor. Before and after each of the sessions, except for the last one, the researchers took pictures of each participant and put the pictures in their personal photo albums. At the beginning of every session (except the first session), the participants were encouraged to look at their own albums to observe their appearances.

Each session of the program included three phases and the participants learned in groups. In the first phase, the instructor invited the participants to recall the learning experiences in the previous session and to share their cosmetic practices at home. In the second phase, the instructor explained and demonstrated the cosmetic activities, and then the participants practiced. Each group was supported by one volunteer or assistant. In the third phase, the instructor guided the participants to observe the changes in their own and their group members’ appearances, to give feedback to one another and to share their afterthoughts on learning.

To encourage the participants to practice what they learned in the beauty program in their daily life, easily accessible beauty products of high quality and low cost, such as lotion that could be bought in supermarkets, were applied. In addition, the instructor made good use of daily necessities, such as porcelain spoons and plastic combs, as massage tools and encouraged the participants to bring and use their own tools. Additionally, to facilitate the participants’ interest in the beauty program, the beauty operation procedures were simplified, and the male participants were provided with multiple alternative activities, such as applying masks instead of putting on make-up and wearing clear lip balms instead of lipsticks. Furthermore, rather than the standard make-up, the importance of personal preferences and experiences of beauty were emphasized.

### 2.4. Data Collection

In this study, the data were collected through questionnaire surveys, semi-structured interviews, and observation.

#### 2.4.1. Questionnaire Survey

Before and after the intervention, the participants were assessed through individual interviews that the researchers administered. These two scales, which have acceptable psychometric properties, have been applied by researchers and clinicians.

*Self-perception of aging* (SPA) among the participants was measured using the Attitudes towards Old People Scale (ATOPS) (α = 0.94), which was developed by Lu and Kao [31] especially for the Taiwanese population. It is a 24-item scale used to assess respondents’ perceptions of and attitudes toward their own aging experiences (e.g., I am old in age but buoyant in spirit). The scale includes 4 subscales: (1) appearance and physical characteristics, i.e., the perception of and attitudes toward one’s aging appearance and body (6 items); (2) psychological and cognitive characteristics (7 items), i.e., the perception of the psychological characteristics and cognitive function in late life; (3) interpersonal relations and social engagement (8 items), i.e., the perception of interpersonal relationships and participation in public activities in late life; and (4) work and economic safety (3 items), i.e., the perception of work and economic safety in late life. Each item is measured on a 4-point Likert scale, from 1 = strongly disagree to 4 = strongly agree, and the total score ranges from 4 to 96. A high score indicates more positive perceptions of and attitudes toward aging experiences.

*Depression* among the participants was measured by the Taiwanese Depression Questionnaire (TDQ) (α = 0.90), a culturally relevant instrument for screening depressive symptoms in Taiwan [32]. TDQ is an 18-item scale that measures the emotion of respondents in the previous week [32] (e.g., “I lost interest in everything”). Respondents measure their emotions on a 4-point Likert scale (0–3), where 0 = never/seldom (the mentioned condition occurred for less than 1 day), 1 = sometimes (the mentioned condition occurred for 1–2 days); 2 = often (the mentioned condition occurred for 3–4 days), and 3 = usually (the mentioned condition occurred for 5–7 days). The total score ranges from 0 to 54. Respondents’ level of depression is assessed as normal (scores ≦ 18), potentially depressed (scores = 19–25), or clinically depressed (scores ≧ 20).

#### 2.4.2. Semi-Structured Interviews

Semi-structured interviews were conducted one week after the beauty program. By interviewing the older adults and care center volunteers, the researchers aimed to understand the learning experiences of the older adults who participated in the beauty program. Using critical case sampling [30], the researchers interviewed the 9 seniors who had participated in activities at the center for a long time and had been familiar with other elderly participants. Additionally, the experienced volunteers who supported the beauty program and had worked in the center for a long time were also interviewed. Information on the interviewed older adults and volunteers is shown in Table 2.

#### 2.4.3. Observation

In this study, participant observation was used to understand the participants’ behaviors and feedback that were relevant to their perception of aging and emotion in the intervention.

### 2.5. Data Analysis

The Statistical Package for the Social Science (SPSS) version 22.0 was used to analyze the questionnaire data. Descriptive statistics were used to describe the participants’ backgrounds. The dependent sample t test was used to examine the changes in SPA and depression scores as the effects of the interventions with a significance level of 0.05.

The interview transcripts and observation records were analyzed by applying thematic analyses [33]. First, the transcripts and records were coded and then categorized into themes. Finally, these themes were refined.

## 3. Results

### 3.1. Results of the Questionnaire Survey

#### 3.1.1. Self-Perception of Aging

As seen in Table 3, the results of the dependent sample *t* tests indicate that the scores of all of the variables of SPA significantly increased from before the intervention to after the intervention (*p* < 0.001).

After the beauty program, the participants’ overall SPA (*t* = −10.37, *p* = 0.000), self-perceptions of appearance and physical characteristics (*t* = −8.10, *p* = 0.000), self-perceptions of psychological and cognitive characteristics (*t* = −8.24, *p* = 0.000), self-perceptions of interpersonal relations and social engagement (*t* = −9.38, *p* = 0.000), and self-perceptions of work and economic safety (*t* = −13.86, *p* = 0.000) all significantly improved. Namely, the participants’ perceptions of and attitudes toward their aging were significantly improved: their late life experiences were more positive.

#### 3.1.2. Depression

As shown in Table 3, the results of the dependent sample *t* tests indicate that the depression scores significantly decreased from before the intervention (M = 7.00, SD = 7.11) to after the intervention (M = 1.59, SD = 1.68) (*p* < 0.001). Namely, after the beauty program, the participants’ level of depression (*t* = 4.32, *p* = 0.000) significantly decreased.

### 3.2. Results of the Interview and Observation Data

#### 3.2.1. Having a More Positive Self-Perception of Aging

The results of the qualitative data analysis indicated that, through the beauty program, the participants’ self-perceptions of aging became more positive in the physical, psychological, and social aspects.

The older adults said that, due to the beauty program, which involved facial cleaning and care, eyebrow shaping, and makeup application, they perceived a significant improvement in their appearance, including looking younger, having clearer skin and fewer wrinkles, becoming beautiful or handsome, and even discovering a different version of themselves. As Elly said, “I never applied makeup before, and I just felt ugly… but after putting on makeup during the beauty program, I was surprised to find myself a different person.” The male participants also noted an improvement in their appearance. “In the beauty program, after the makeup application, I found the dark spots on my face had faded, and my friends said that I looked like a teenager,” Dick said. The older adults also found themselves more energetic after practicing skincare and applying makeup. “After taking the beauty program, the older adults looked much better than before and became more enthusiastic about beauty, like lipstick application or eyebrow penciling,” the volunteer Jenny noted.

The beauty program encouraged the participants to observe their appearance, which accordingly led them to appreciate themselves after applying makeup. At the beginning, the female participants mostly failed to accept their own appearance and thus disliked taking photos or viewing their own photos. However, during the beauty program, in addition to guiding the participants in cleaning and caring for their faces and applying makeup, the instructor showed the participants photos of themselves before and after the makeup application to encourage them to observe themselves and each other, give mutual positive feedback, and gradually accept and even appreciate their appearance. As Candy said, “I’m not going to take my makeup off after class today. Rather, I will leave this makeup on when going home and show it to my neighbors and family members. I feel pretty after the makeup, rather than looking like an old woman in her 80s.” Ida also noted, “I appreciate myself with light makeup. The more I look at myself with makeup, the more I like myself”.

In the beauty program, most of the participants felt that they were more outgoing and willing to help each other in the group. As Yuki said, “Lyn’s son recently passed away. While learning about skincare and massage in class, we talked and laughed together hoping to comfort and encourage Lyn.” The volunteer Kyle noted, “Although they have been neighbors for a long time, these older participants are shy to serve each other. However, mutual massage in this beauty program can gradually make them open to massaging each other.”

#### 3.2.2. Feeling Pleasant and Relaxed through Skincare, Makeup Application, Aromatherapy, and Mutual Positive Feedback

The older adults and volunteers said that the beauty program had direct positive effects on the moods of the older adults by improving their appearances. Most of the participants said that they became happy after applying makeup. In addition to looking at themselves in the mirror to identify changes in their appearance, they also felt cheerful when receiving positive feedback from their partners. As Betty mentioned, “I never applied makeup before, but after this program, I realized that applying makeup or not makes a world of difference. I like my appearance with light makeup. When going out, I really feel joyful when I am praised for becoming more beautiful”.

Moreover, the use of aromatherapy together with essential oil massage relaxed the participants by smelling and massaging with essential oils. Hebe said, “Essential oil massage, together with hot compresses, can unwind my body, especially the shoulder and neck, making me in a good mood”.

#### 3.2.3. Promotion of Social Interaction with Friends and Family

The participants attending the beauty program drew more attention from their neighbors, friends, and family, which generated more interaction among them. As Apple mentioned, “When I got home with makeup from the beauty program session, my friends and neighbors usually asked me where I went and why I applied such exquisite makeup, which always led to a lively conversation about my makeup and the beauty program.” Additionally, the participants’ families, such as their spouses, children, and grandchildren, also noticed the changes in their appearances when the participants returned home after the beauty program session. Furthermore, some of them uploaded their photos taken during the beauty program to social media, eliciting warm and positive responses from their family. Dick said, “My son usually told me that I looked more handsome and younger and often discussed my learning in the care center! Before, he seldom paid attention to my activity”.

#### 3.2.4. Disrupting Stereotypes about Makeup

It was found that older females in the agricultural community, which is characterized by a conservative social atmosphere, were not encouraged to practice skincare or apply makeup when they were young, “Because we had been taught to keep our look as simple as possible and to be busy farming or doing work instead of dressing up ourselves,” Ida said. Hence, they tended to believe that makeup was necessary only for important activities and worried that makeup would cause others to think they were too flashy. As the researcher observed, “Jade requested that her eyebrows be returned to normal immediately after eyebrow penciling, even though she admitted that her eyes looked brighter with eyebrow penciling. This is because she was not accustomed to eyebrow penciling and was afraid of being criticized by her husband or laughed at by neighbors.” However, in the beauty program, the participants were encouraged to implement facial care, put on makeup, and observe the continued improvement of their facial appearance. Furthermore, the instructor continued to reinforce the idea that regardless of their age, the individuals should accept and care for their appearance and that applying makeup can prevent dementia [3,26,27]. Gradually, the participants began to dress up for themselves in a natural way, accepted that maintaining and improving their appearance was normal, and stopped worrying about others’ comments. As Green said, “At the beginning, I was embarrassed when meeting people after applying makeup. I was worried that people would wonder why I suddenly applied makeup for no reason, and I felt like I did something bad. However, because in the beauty class, we always applied makeup together naturally, I’m getting used to putting on makeup”.

During the implementation of the beauty program, even the three male participants engaged in the beauty activities. Dick said that the facial care and makeup activities made him happy and, without being restricted by the traditional stereotype that makeup is only for females, he was willing to participate in the beauty program. Furthermore, some female elderly participants even encouraged the male older adults to attend the beauty program. Betty noted, “I believe the men will also feel happy when learning to apply makeup, including putting lipstick on their lips”.

Overall, the beauty program in this study could improve self-perception of aging and alleviate depression among the older adults living in a rural area. Moreover, through the beauty program, the older adults could better accept and appreciate their own appearances, thus contributing to the improvement of their social lives. Both the male and female participants managed to disrupt stereotypes about makeup and viewed maintaining and improving their appearance as being normal and natural.

## 4. Discussion

In this study, the quantitative results show that the participants’ overall self-perceptions of aging were significantly improved after the beauty program, and the qualitative results further indicated that the program could promote their self-perceptions of aging in the physical, psychological, and social aspects. Specifically, the participants believed that their appearance was significantly improved through the beauty program, including by looking younger, more beautiful, and elegant. In fact, the improvement of physical appearance is essential for older women’s quality of life. Research has suggested that female older adults who perceive themselves as being physically attractive and young tend to feel healthy, happy, optimistic, and socially connected; to adjust to the environment well; to be satisfied with life; and to be willing to attend social activities [34,35,36].

The beauty program in this study, which emphasized facial cleaning and care, makeup application, and aromatherapy, and which was implemented in groups, was likely to have impacts on physical, psychological, and social aspects of the participants’ aging experiences. However, the results of the questionnaire surveys indicate that the older adults’ perceptions of aging in terms of job and economic security were also improved. Clearly, their perceptions of their work and economic security were not only affected by objective life indicators but also closely related to their psychological feelings [5,37]. Hence, the beauty program, which directly improved the participants’ physical, psychological, and social perceptions of aging, could indirectly facilitate their attitudes toward their current work and economic security.

The quantitative results of this study indicate that the beauty program significantly alleviated depression among the participants, and the qualitative results also show that the participants felt happier and more relaxed after participating in the beauty program. This may be explained by the fact that positive peer interaction in the beauty program, positive feedback from friends and family members, and active social engagement in the beauty program could improve the psychological health of the older participants. Moreover, the improvement in their physical appearances may also have had positive psychological effects on the participants’ emotions. Clinical studies have found that beauty programs can effectively improve the mental health of patients with cancer, dementia, or psychological illnesses [3].

Empowering older adults is one of the key objectives of health promotion programs for the aging population. Namely, older adults are encouraged to realize their potential and gradually become confident and independent through continuous learning [1,13]. In this study, it was found that the participants could become empowered by achieving self-improvement through self-practice and learning beauty techniques, and by improving their appearances with continuous facial care and makeup. When initially designing the program, the researchers worried about the ability of the older adults to independently pencil their eyebrows, as eyebrow penciling requires strong eyesight and subtle hand–eye coordination. In the beauty program, when penciling their eyebrows by themselves for the first time, the participants were obviously diffident and required assistance and frequent confirmation about their performance. However, in the second eyebrow penciling session, one of the researchers observed, “most of the participants, like Apple, Candy, Yuki, Lyn, Snow, Lily, May, Dick, are even able to apply makeup themselves without relying on the assistants. Their self-confidence in their ability to apply makeup seems to be increasing.” In subsequent sessions, the participants rapidly completed eyebrow penciling in a more independent and confident manner. Additionally, Green noted, “This program is great for me. At first, I felt embarrassed to apply makeup because I seldom did that. But now, before going out, I’m used to practicing basic facial care and applying makeup based on what I learned in the beauty program, and I am becoming increasingly proficient in doing so. My granddaughter-in-law invited me to take family photos next week, and I will apply makeup for it.” All of these findings reveal the older adults’ strong ability and motivation to learn, and their facilitated sense of self-efficacy. In the process of aging, people may experience a variety of dramatic physical, psychological, and social changes. The improvement in older adults’ senses of self-efficacy can stimulate their ability to control their environment, reduce the negative impact of environmental changes, and allow them to positively adjust their self-psychology, thus avoiding the sense of helplessness and depression [8].

Furthermore, maintaining and pursuing an attractive appearance plays a key role in the later lives of older adults because having an optimal appearance indicates that older adults can effectively manage their aging bodies. Additionally, positive feedback from themselves and others about their physical appearances can boost their self-confidence and then contribute to their empowerment [36].

This study reveals that the beauty program could provide older adults with new learning experiences and lead them to find a different version of themselves and further appreciate themselves. Elderly people in rural areas may rarely apply facial makeup or practice aromatherapy in their lifetime due to their heavy workload or the conservative social atmosphere. As Hebe noted, “In my lifetime, I only applied makeup when getting married, so it was not until at this age that I learned applying makeup at the center.” Additionally, the participants could see a different version of themselves through applying makeup in the beauty program. “I can hardly imagine how beautiful I could be after applying makeup,” Elly said. This was an unexpected result of the beauty program. Most participants who seldom wore makeup were so surprised at the improvement in their appearance that they could not help but repeatedly observe their faces in a mirror for a long time after applying makeup. However, this finding has rarely been mentioned in previous studies [3]. This may be explained by the fact that previous studies have focused on the female aging population living in urban areas, most of whom may be accustomed to make-up [38], while in this study, the female older adults lived in agricultural areas and rarely used makeup. Moreover, most of the previous studies have adopted a quasiexperimental design and collected data utilizing questionnaire surveys, while this study adopted the mixed methods approach and gathered multiple types of data using questionnaire surveys, interviews, and observations. Only by collecting comprehensive data using the mixed methods approach can the authentic and comprehensive lived experiences of older adults in beauty programs be extracted and understood [29].

The study also found that the beauty program could reverse the female participants’ negative body images and help them maintain and pursue an attractive appearance. At the beginning of the program, most of the female participants had negative self-appraisals of their appearances. This may be explained by the traditional Chinese social practice of replacing modesty with self-deprecation, which is especially common in rural areas [39]. Meanwhile, the aging population tends to have a more negative body image than young adults. Notably, the belittlement of their appearances can damage older female adults’ self-concept [22]. Moreover, this study found that, in the beginning of the study, most of the female participants were often worried about being criticized for pursuing and maintaining an attractive appearance. Such fears are especially common in rural areas, where people are more conservative and may restrict the pursuit and maintenance of an attractive appearance, thus negatively affecting individuals’ self-identities [40]. This phenomenon might be explained by the collectivistic cultures and social orientation of behaviors among the Chinese, who tend to value others’ appraisal, especially older Chinese women [39]. This context is different from the beauty culture among females in South Korean metro areas, for whom daily makeup practices mean “designing aging” rather than “denying aging.” For women in South Korean metro areas, to please themselves while pleasing others, it is necessary to maintain a natural and age-appropriate appearance, and individuals’ self-confidence may be boosted by positive feedback from others. Therefore, for these women, maintaining an appropriate appearance is the intersubjective social encounter [38].

It was found that the beauty program in this study could disrupt the older adults’ stereotype that “makeup is only for females” and attract the older men to participate in beauty activities. In the beauty program, there were three male participants who actively and willingly participated in beauty activities, such as applying makeup, lipstick, and facial masks, as well as practicing essential oil massage, and warmly praised the group members’ appearances after practicing skincare and applying makeup. One of the male participants, Dick, always left the center with applied red lipstick; he said, “I would like to show my good looks to my friends and family.” He usually encouraged his older male friends to visit the beauty program. Additionally, in the end, even the female participants suggested that males should also attend the beauty program. This finding could be explained by the fact that the beauty program provided varied activities for the male older adults to choose from to achieve similar skincare effects [19,22]. For example, the instructor also offered facial mask application to the male participants in the makeup session, offered clear lip balm as a substitute for colorful lipsticks, and offered body lotion for massage in the essential oil massage activity. Moreover, encouragement and positive feedback from female peers may also encourage the older men to participate in beauty activities.

## 5. Limitations

It is important to acknowledge some limitations of this study. First, the participants consisted of elderly adults in good health, who may be different from the general aging population in some respects. Therefore, the generalizability of the results is limited. Second, the beauty program in this study was multidimensional in nature; therefore, we can tentatively conclude that the effects of self-perception of aging and depression resulted from the cosmetic treatments. Third, a small number of the participants withdrew from the study, and they might have had different reactions toward the beauty program from the participants who completed the program. These differences might have influenced the results and warrant further examination. Fourth, the beauty program in this study comprised three parts, including skin care, make-up application, and massage with essential oils, and the specific effects of each part of the beauty program warrant further investigation.

## 6. Conclusions

The results revealed that the beauty program in this study was effective for improving the self-perceptions of aging and reducing depression among the community-dwelling older adults in an agricultural area in Taiwan. Additionally, the older participants’ body images were improved, the participants disrupted their stereotypes about makeup, and they were willing to gradually maintain their appearance. Further research with a larger population of aging people, male older adults, or frail older people is needed to examine the specific effects of the beauty program.

## Figures and Tables

**Table 1 healthcare-11-01377-t001:** Protocol of the Beauty Program.

Session No/Topic	Themes
1. Facial cleansing	* Greeting * Introduction to the beauty program * Basic facial care: cleaning with makeup-removing cotton pads, moisturizing with lotion, and applying sunscreen
2. Dewy skin	* Basic facial care * Facial massage: facial self-massaging with massage cream, rinse-off mask, and warm facial compress with towels
3. Shoulder and neck massage	* Basic facial care * Shoulder and neck massage: massaging with lotion in pairs and warm compress with towels
4. Arm and hand massage	* Basic facial care * Arm and hand massage: massaging with lotion in pairs and warm compress with towels
5. Eyebrow shaping I	* Basic facial care * Eyebrows shaping: trimming eyebrows by the instructor, drawing eyebrows with eyebrow brushes
6. Eyebrow shaping II	* Before and after drawing eyebrows: examination of the 3 female participants’ photos of before and after drawing eyebrows * Basic facial care * Eyebrow shaping: drawing eyebrows with eyebrow pencils * Hand massage: massaging with lotion in pairs and warm compress with towels
7. Eyebrow shaping III	* Basic facial care * Eyebrow shaping: drawing eyebrows with eyebrow pencils * Neck and shoulder massage: massaging with lotion in pairs and warm compress with towels
8. Essential oil massage I	* Aroma inhalation: inhalation of essential oils, introduction of essential oils * Neck and shoulder massage: massaging with essential oils in pairs * Arm and hand massage: massaging with essential oils in pairs * Warm compress with towels
9. Essential oil massage II	* Aroma inhalation * Ear massage: massaging with essential oils in pairs * Head massage: massaging with essential oils in pairs * Warm compress with towels
10. Make-up I	* Before and after make-up: examination of the 3 female participants’ photos before and after applying make-up * Basic facial care * Make-up: applying loose powder, blush, and lipstick and drawing eyebrows with eyebrow pencils
11. Make-up II	* Basic facial care * Make-up
12. Make-up III	* Basic facial care: cleaning with makeup-removing cotton pads, moisturizing with lotion, and applying sunscreen * Make-up * Body massage: massaging with essential oils in pairs * Warm compress with towels
13. Make-up IV & Looking back	* Basic facial care * Make-up * Looking back: looking at photos of the previous 12 sessions * Feedback: the participants’ feedback on the program

**Table 2 healthcare-11-01377-t002:** Information of interviewees.

Name	Identity	Gender	Age
Apple	elderly individual	female	74
Betty			75
Candy			85
Elly			78
Yuki			76
Green			65
Hebe			87
Ida			74
Dick		male	83
Jenny	volunteer	female	61
Kyle			64

**Table 3 healthcare-11-01377-t003:** Differences in self-perception of aging and depression between pretest and posttest (*n* = 29).

Variables	Pretest		Post-Test		*t*
	M	S.D.	M	S.D.	
**Self-perception of aging** ^a^	69.41	6.40	89.79	7.95	−10.37 ***
Appearance and physical characteristics ^a^	16.38	1.99	21.21	2.82	−8.10 ***
Psychological and cognitive characteristics ^a^	21.24	2.59	26.69	2.21	−8.24 ***
Interpersonal relations and social engagement ^a^	24.03	2.54	30.66	2.61	−9.38 ***
Work and economic safety ^a^	7.76	0.79	11.24	1.24	−13.86 ***
**Depression** ^b^	7.00	7.11	1.59	1.68	4.32 ***

Note. *** *p* < 0.001; ^a^ a higher score indicates a better situation; ^b^ a higher score indicates a worse situation.

## Data Availability

The data are not publicly available due to the privacy of the participants.

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
