# Peer review of "The Effects of a Beauty Program on Self-Perception of Aging and Depression among Community-Dwelling Older Adults in an Agricultural Area in Taiwan"

_healthcare, 2023, doi:10.3390/healthcare11101377_

Round 1
Reviewer 1 Report
Dear Authors, thank you for submitting this manuscript. It is very important to identify the effect of a beauty program on the self-perception of aging and depression in the elderly. I leave here my considerations of the text.
1. line 37 - "unable to live normally" review that statement
2. line 45 - develop the 'agism' concept
3. line 72 - it is necessary to include more references, only one was inserted as an example.
4. line 72-91 - here is the main part of the study, I suggest reducing the previous paragraphs and developing this subject further. For example, bring more information about the studies cited in the next paragraph (e.g., Hayakawa, et al., 2016; Machida, et al., 2012; Nakano, et al., 2019ï¼›Ryu & Lee, 2021; Sakatani, 2015) .
5. line 111 - this sentence seems to be loose here (referring to health professionals).
6. line 124-138 - I can understand everything in the paragraph, however there is a lot of different information, from ethics committee approval, sample characteristics and adherence to the program. Try to improve the layout of information.
7. "Research design" - Research design? the research was applied, so it is not a project, review the title.
8. I have doubts about the case study, are you describing the entire intervention, wouldn't it be a quasi-experimental study?
9. line 152 - sentence seems incomplete, revise.
10. line 152-162 - the paragraph is repeating and confusing, I suggest revising this part.
table 1 - replace second column marker (that circle is huge)
11. line 172 - "warm-up, main activity, and wrap-up," these terms resemble physical training, I suggest opting for other terms, for example (phase or moment 1, 2 and 3 of the session).
12. item 2.4 - this has been said before, check which location is better.
13. check need to present table 2.
14. line 299- 311 - the report is beautiful, however it reminds me of a quest to 'hide' the marks of age (I believe this will be discussed later on).
15. "Before, he rarely paid attention to my 344 activity.”" How cool is this passage, an excellent example of getting closer to the family.
16. "and that applying makeup can prevent dementia" does this statement have any reference?
17. line 389-394 - the statement refers to older women, I suggest revising it because there are men in your study. Or is this statement from the cited studies?
18. I realized that the focus of the study is more on makeup, but there were other moments as important as, for example, self-massage and aromatherapy. Why were these interventions not discussed in the interviews?
19. line 455-456 - I beg your attention here, as mentioned in item 14, I am concerned if they were hiding age marks. So far it has not been discussed.
20. It would be interesting to look for seniors who dropped out and really understand why they did not continue in the program.
21. line 522- remove this section "Despite these limitations,".
22. "Additionally, the female older participants’" - remember that there are men in the survey.
Small English language edition required
Reviewer 2 Report
Thank you very much for offering me the opportunity to review this very informative and interesting manuscript. The manuscript describes the outcomes of a beauty program on self-perception and depression in an aging population in Taiwan. The authors provided excellent explanations regarding the importance of this study. Additionally, they provided clear implications of the outcomes. I have only very minor suggestions for improving the introduction and methodology sections of the manuscript. The manuscript is in general very well written.
Introduction:
· 3rd paragraph (line 66-68), sentence starting with “The suicide rate of older people is twice...” needs a reference.
· 4th paragraph (line 83-85) sentence starting with “Studies have also confirmed...” needs references.
· 4th paragraph (line 92) – a full-stop is missing at the end of the paragraph.
· 5th paragraph (line 98) – should be “have not been clearly presented” and not “has not been”
Methods:
2.4.1 Questionnaire survey
· The questionnaire section would benefit from additional information on the psychometric properties of the two scales used and a clarification if the alpha-values reported are from previous studies or from this study.
· Psychometric properties reported from previous studies require references (e.g. in line 197).
· The statistical analyses paragraph could benefit from having its own subsection and from additional information if normality tests were conducted before deciding on the statistical analysis used.
The use of the English language within the manuscript is of very high quality. Only one minor change is needed, which is indicated in the comment section above.
Reviewer 3 Report
The review of the article “The Effects of a Beauty Program on Self-perception of Aging and Depression among Community-dwelling Older Adults in an Agricultural Area in Taiwan”
The aim of the paper was to investigate the effect of a beauty program on self-perception of aging and depression among the community-dwelling older adults in an agricultural area in Taiwan. The authors’ approach in applying multiple and qualitative research methods and understanding the daily lived experiences of health and illness in real sociocultural contexts is an example of comprehensive bio-psycho-social approach to research of health and perception of health. Such approach is crucial for better understanding of the quality of life related to health and development of the evidence-based treatment options. The manuscript is clear, relevant for the field and presented in a well-structured manner.
1. The title “The Effects of a Beauty Program on Self-perception of Aging and Depression among Community-dwelling Older Adults in an Agricultural Area in Taiwan” accurately reflects the content.
2. The abstract is structured, concise and specify outcome measures.
3. Introduction justify performing the study.
4. The sample and its formation described in detail, the inclusion and exclusion criteria are stated.
5. The methods are they supported by references.
6. The results are clear and convincing with abundant qualitative references.
7. Discussion begins with the most important findings and relates exclusively to the results of the study, the limitations of the study are clearly stated.
8. Conclusions are they based on the presented results.
9. References are they accurate and up-to-date (without an excessive number of self-citations and relevant mostly recent publications)
